# GAC 2021 Proposal

**Title:** What makes representations "useful"?

**Scientific question:** Internal representations play a central role in the study of both biological and artificial intelligence, as well as in philosophy of mind, but what precisely defines a representation is challenging to pin down. Across disciplines, one common thread is that representations are typically "useful" in some sense. Centering around this concept of usefulness, we propose a cross-disciplinary GAC to share ideas and develop more precise answers to the following questions:

1. What makes representations "useful," both in terms of their *content* and their *form*?
2. How does the *use* or downstream causal effect of a representation contribute to its meaning?

We will simplify the scope by primarily discussing these questions in the context of visual perception.

**Background:** Representations play a central role in cognitive science, neuroscience, philosophy of mind, and machine learning. Painting in broad strokes, "representation" in philosophy often goes hand-in-hand with intentionality, or the idea that minds manipulate *meaningful* symbols, and meaningful representations play an explanatory role in phenomenology. In cognitive science and artificial intelligence, the idea that an agent internalizes representations of the world – in terms of entities (like objects) and structures (like maps) – helps to both explain and design intelligent systems. In machine learning, designing or learning the right representations can significantly impact the efficiency of downstream tasks including both further learning and inference. While some pluralism of ideas is to be expected, there is also much to gain by exploring connections between these different aspects of representation across disciplines.

Researchers in each of these fields are often inspired by ideas or developments in the others, and motivated by some of the same basic questions, such as those surrounding representations. Disagreement and misunderstanding about the term "representation" has the potential to hinder scientific communication and duplicate research efforts [6, 11, 59, 31, 25]. If the main source of confusion is a plurality of serviceable notions of representations, our GAC will help to clarify what these notions are and in which contexts they are appropriate. If, indeed, a deeper confusion about representations stymies innovative research, then our GAC presents an opportunity to clarify and refocus research in multiple fields, and to cultivate cross-disciplinary researchers.

Ideas about "useful" representations are seeing a resurgence in different forms, making it a timely and exciting subtopic for collaboration across disciplines. We will next summarize some active areas related to "usefulness" in each discipline.

An active area in **philosophy** examines *embodied* cognition and how cognitive and perceptual processes might involve *affordances*, or possibilities for action in the perceiver's environment. It remains unresolved what the implications of embodiment and affordances are for our understanding of representations [47, 54, 27, 3, 50, 52, 49, 19]. Further, there is a (re)animated literature around how *teleology* – roughly, functions or purposes derived from the process of evolution – should inform our understanding of representations and mechanisms in neural systems [15, 4, 53, 26, 14, 41].

In **neuroscience**, "usefulness" appears in many guises. For one, many have pointed out the insufficiency of correlational methods for making representational claims; equally important is how neural activity influences downstream functions, or put another way, how the hypothesized representation is actually *used* [46, 45, 11, 6]. A related problem is leveraging animals' natural behaviors and ecological niches as a way to constrain theories of neural representation [20, 18, 42]. Other senses of "usefulness" derive from optimization for a task or set of tasks [39, 61, 32, 58], or task-independent

prediction of past or future inputs [43, 44, 16, 55]; learning an internal model of the world may be a possible middle-ground [34, 9].

There have been many exciting recent developments in **machine learning** on topics related to representation-learning from unlabeled and minimally structured data. As a result, there is a growing and evolving set of quantitative metrics for what makes a model or representation useful. One area is elaborating classic concepts from Information Theory (IT) to explain the success of deep learning [57, 1, 23], and to address technical limitations that have made IT impractical for studying representations [60]. There have also been recent developments in defining "disentangled" representations [8, 29, 48, 24], along with new design patterns and training objectives to achieve them by unsupervised methods [28, 13, 2, 35]. New ideas to self-supervised learning, learning from structured data like video, and active/causal perception has led to further breakthroughs in representation learning [17, 36, 21, 56, 40, 51]. Another active area in deep learning is uncertainty quantification, where there is renewed interest in questions like *what* types of uncertainty to represent and *why*, and various proposals for how to do so in practice [30, 10, 33, 12][1].

The dominant tools for quantitatively studying representations in machine learning and neuroscience incorporate varying senses of *usefulness* and *meaningfulness*, which are crucial in philosophical accounts of representation. For example, Information Theory and Mutual Information have been extensively applied throughout neuro- and cognitive science [5, 7, 22] as well as in deep learning [30, 57, 1]. However, it has been pointed out that this kind of information is not necessarily "usable" in principle [60, 23] or "used" in practice [45, 6]. Another popular suite of tools compares representations based on their geometry [38, 37], which identifies "useful" representations insofar as the same information is relevant for two systems (e.g. brains and neural networks). Further, these methods are sensitive to statistical dependencies between representational spaces, but not to their downstream causal effects.

**Challenges, competing hypotheses, and proposed approach for resolution:** The primary challenge will be identifying specific areas of overlap across fields, given their diverse ideas and vocabularies. We will therefore structure the GAC in two parts: the first will be a brief set of tutorials in which representatives from philosophy, machine learning, and neuroscience will each give an introduction to concepts, seminal studies, operational definitions, and open questions about "useful" representations in their respective fields. In the second part of the GAC, we will hold moderated discussions and debates on specific topics structured around the high-level "scientific questions" outlined at the beginning of this document. These high-level questions are paraphrased in bold below, alongside possible fine-grained discussion topics. This format and the questions below are subject to changes given feedback from the community.

1. **What makes representations "useful" in terms of their *content* and their *form*, and how are these related?**

   **What to represent (content):** To what extent is it task performance or reward all the way down versus task-free model-building all the way up, and how do these interact? How do embodiment, affordances, and ecology further shape what is useful to represent? In what ways is it useful to represent uncertainty? In what ways is it useful to represent causal relations?

   **How to represent it (form):** What are the principles of designing usable representations – things like disentanglement, independence, invariance, equivariance, efficiency, smoothness,

---

[1]Representing uncertainty is also a contentious topic in neuroscience and cognitive science, with two of last year's GACs devoted to questions on representation of probability in the brain.

and decodability – and how do they relate to each other? How do these interact with representational content, if at all? What existing evidence is there for each of these properties in the brain, or what new experiments are needed to answer this?

2. **The role of actual and potential *use*.** Are different questions answered by knowing the potential uses of a representation versus knowing the actual downstream effect that a particular representation has? What are the conceptual, experimental, or technical barriers to quantifying causal interactions among internal representations, or between internal representations and behavior? How do salient actions and threats for an organism broadly influence the representations it uses? How, if at all, do evolutionary histories constrain biological representations, and does something play an analogous role for artificial ones?

**Concrete outcomes:**

- A taxonomy of useful forms of representations across disciplines, and how they relate.
- A set of empirically testable questions – and experimental methodologies – about representational forms in biological neural systems.
- Updated philosophical theories of representation, and mathematical tools for quantifying representations and representational similarity, taking into account emerging ideas in ML on disentanglement, causality, etc.

**Benefit to the community:**

- Our collaboration will allow for deeper and broader insight into a far-reaching set of questions than an investigation from within any one discipline is likely to achieve. It will also serve to cross-pollinate ideas between disciplines, and to facilitate future inter-disciplinary collaborations.
- We will strongly encourage didacticism and open-access materials for all participants so that tutorials and discussions will serve as an ongoing teaching resource.
- We will draw from diverse backgrounds and career levels for the participants, creating an impactful and rare opportunity for researchers that face unique challenges, and serving as a model for inclusive science.

**Core members:** The final team is subject to change, and we welcome new ideas and new members from the community. If this proposal is accepted, the following members have all initially agreed to help organize or advise the GAC, to co-author a summary paper, and to share updates at CCN 2022.

| Name | Position/Institution | Expertise | Role |
|------|---------------------|-----------|------|
| Ben Baker | Postdoc/UPenn | Philo | Organizer |
| Richard Lange | Postdoc/UPenn | Neuro (ML) | Organizer |
| Alessandro Achille | Applied Scientist/Amazon | ML | Organizer |
| Rosa Cao | Professor/Stanford | Philo (Neuro) | Organizer |
| Niko Kriegeskorte | Professor/Columbia | Neuro (Philo) | Advisor |
| Odelia Schwartz | Professor/Miami | Neuro | Advisor |
| Xaq Pitkow | Professor/BCM/Rice | Neuro (ML) | Advisor |

**Signed:** Ben Baker, Richard Lange, Alessandro Achille[2], Rosa Cao, Nikolaus Kriegeskorte, Odelia Schwartz, Xaq Pitkow

---

[2]Paper co-authorship pending approval from Amazon.

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
