# OpenReview forum: "What makes representations “useful”?"
_ccneuro.org/CCN/2021/Workshop/GAC_

### Official Review · ~Adrien_Doerig1 · 2021-07-23
**Interesting and important multidisciplinary topic, loosely defined methodology**

**Rating:** 7
**Confidence:** 4

**Review:**

Baker et al. propose a GAC to clarify the notion of representation. This is an important topic since, as the authors point out, "representation" is an important concept across multiple disciplines that currently lacks a consensual meaning.

To remedy the situation, Baker et al. propose set of tutorials at CCN, followed by moderated discussions and debates on related to the question at hand.

Pinning down what should be meant by "representation" is a very relevant and timely debate, which will greatly benefit from an expansive and inclusive conversation amongst researchers. For this reason, I find this proposal compelling and I suggest accepting it.

One limitation is that this proposal does not seem to fit exactly in the generative *adversarial* collaboration, since there are not two well defined camps that propose to adversarially collaborate to resolve a question using a well-defined approach. Instead, it seems more like an exploratory collaboration. There is a slight concern that this may make the collaboration less adversarial and therefore a little off topic for this "venue", and perhaps a little less effective. However, I am not very familiar with this GAC format yet, so I may be wrong.

---

### Official Review · ~SiQi_Zhou1 · 2021-07-26
**Interesting and relevant topic for GAC**

**Rating:** 9
**Confidence:** 3

**Review:**

The authors proposed a GAC focusing on quantifying the quality of "representation" in the context of multiple research domains (i.e., philosophy, neuroscience, machine learning). The GAC topics/questions are interesting and relevant, and the proposed GAC is also structured to facilitate interdisciplinary discussions. The GAC would also benefit from the diversity of the expertise of the organizers.

One minor thought I had was that while "representation" is relevant across the three aforementioned research fields, I wonder if there is a common application/benchmarking problem that would make sense for debating/comparing the "usefulness" or "meaningfulness" of representation. This may help to reach a more qualitative conclusion from the interdisciplinary discussion.

---

### Official Review · ~Arthur_Prat-Carrabin1 · 2021-07-26
**Interesting and important questions**

**Rating:** 9
**Confidence:** 4

**Review:**

This proposal aims at exploring the notion of "usefulness" of internal representations. The authors argue that although philosophy, neuroscience, and machine learning have adopted different approaches to the question of the nature of representations, ideas relating to the importance of the "usefulness" of representations have emerged in these three fields, under different guises. The authors thus propose a two-part GAC, with a first part dedicated to tutorials and to the presentation of the main concepts and problems in each field, and a second part including discussions articulated around two main questions: "What makes representations “useful,” both in terms of their content and their form?" and "How does the use or downstream causal effect of a representation contribute to its meaning?".

The proposal adresses, in my opinion, an important topic, around which some clarifications about different approaches and concepts would be much welcome; and one can hope that bringing together people from different disciplines will cross-fertilize ideas and be beneficial for all. As for the proposed format, I don't know if it fits exactly what a GAC is supposed to be, but it seems appropriate for the goals of the authors.

I find the prospect of such an event to be exciting, and thus I am in favor of accepting the proposal. I have, however, a number of remarks:

- First, it seems to me that the notion of "cost" and/or "constraint" is missing in the presentation of the topic. One representation might be more "useful" than another, but it might also be very costly, or even impossible, to implement. (For instance, remembering all the visual scenes observed in one's life would probably be useful, but the cost would certainly be prohibitive.) This aspect of the problem is crucial because the content and form of the actual representations in the brain are likely to depend not only on how useful they are, but also on how costly they are, and on whether they are feasible at all. What kind of costs and/or constraints weigh on the representations – e.g., on their complexity, on their precision, or on the memory load they entail –, and on the computations upon them? In my opinion this question comes hand in hand with that of the "usefulness" of representations.

- Second, isn't it somewhat regrettable that the authors choose to focus on "visual perception"? The questions asked are relevant in other contexts, and it would be a shame to do without insightful contributions just because they relate to representations unrelated to vision.

- Third, the scope of the second question is not very clear to me; it seems that it is a sub-question of the first one.

---

### Official Review · ~Xiaoxuan_Jia1 · 2021-07-26
**What to represent, how to represent and the 'usefulness'**

**Rating:** 9
**Confidence:** 5

**Review:**

The proposal raised an interesting issue in the field: the definition of 'representation'. When people from different fields, like Neuroscience, deep learning and machine learning, use the same word, what do they refer to exactly? The proposal raised a follow up question: the definition of 'usefulness' of a representation. I think the meaning of representation is related to how we phrase the scientific question. As suggested in the proposal, the discussion will focus on what to be represented, how to represent and finally how to define 'usefulness' based on the purpose.

The proposal is clear in terms of the question to discuss (and potentially debate) and raised an important issue. The hope is to bring people on the same page or start to think deeper when using these words (better with caution). I think this will be an interesting topic for this conference to discuss and may inspire collaborations across fields.